# Identity and COVID-19 in Canada: Gender, ethnicity, and minority status

**Roland Pongou[1], Bright Opoku Ahinkorah[2], Marie Christelle Mabeu[3], Arunika Agarwal[4], Stéphanie Maltais[5,6], Aissata Boubacar Moumouni[1], Sanni Yaya[5,7] ***

1 Department of Economics, Faculty of Social Sciences, University of Ottawa, Ottawa, Ontario, Canada,
2 School of Public Health, Faculty of Health, University of Technology Sydney, Ultimo, NSW, Australia,
3 Department of Economics, Stanford University, Stanford, California, United States of America,
4 Department of Global Health and Population, Harvard T.H. Chan School of Public Health, Boston, MA, United States of America, 5 School of International Development and Global Studies, Faculty of Social Sciences, University of Ottawa, Ottawa, Ontario, Canada, 6 The Public Health Agency of Canada (PHAC), Ottawa, Ontario, Canada, 7 The George Institute for Global Health, Imperial College London, London, United Kingdom

* sanni.yaya@uOttawa.ca

## Abstract

### Background

During the COVID-19 pandemic, growing evidence from the United States, the United Kingdom, and China has demonstrated the unequal social and economic burden of this health crisis. Yet, in Canada, studies assessing the socioeconomic and demographic determinants of COVID-19, and how these determinants vary by gender and ethnic minority status, remain scarce. As new strains of COVID-19 emerge, it is important to understand the disparities to be able to initiate policies and interventions that target and prioritise the most at-risk sub-populations.

### Aim

The objective of this study is to assess the socioeconomic and demographic factors associated with COVID-19-related symptoms in Canada, and how these determinants vary by identity factors including gender and visible minority status.

### Methods

We implemented an online survey and collected a nationally representative sample of 2,829 individual responses. The original data collected via the SurveyMonkey platform were analysed using a cross-sectional study. The outcome variables were COVID-19-related symptoms among respondents and their household members. The exposure variables were socioeconomic and demographic factors including gender and ethnicity as well as age, province, minority status, level of education, total annual income in 2019, and number of household members. Descriptive statistics, chi-square tests, and multivariable logistic regression analyses were performed to test the associations. The results were presented as adjusted odds ratios (aORs) at p < 0.05 and a 95% confidence interval.

**Data Availability Statement:** The survey data cannot be shared publicly as they hold potentially attributable sensitive information regarding the participants. It would therefore be unethical to

make them public and would undermine the ethical committee agreement and consent process. Data can be requested to the University of Ottawa Office of Research Ethics and Integrity by researchers who meet the criteria for access to confidential data. Office of Research Ethics and Integrity Tabaret Hall 550 Cumberland St Room 154 Ottawa, ON, Canada K1N 6N5 Tel.: 613-562-5387 Fax.: 613-562-5338 ethics@uottawa.ca All other relevant data are presented within the article.

**Funding:** This work was supported by the Social Sciences and Humanities Research Council of Canada's Partnership Engage Grant # 231377-190299-2001 (RP) and the Social Sciences and Humanities Research Council of Canada's Insight Grants # 231415-190299-2001 (RP). The funders had no role in study design, data collection and analysis, decision to publish, or preparation of the manuscript.

**Competing interests:** The authors have declared that no competing interests exist. The work was not undertaken under the auspices of PHAC as part of employment responsibilities. It was conducted under the author's other affiliation and any views expressed therein are personal opinions and not those of PHAC.

## Results

We found that the odds of having COVID-19-related symptoms were higher among respondents who belong to mixed race [aOR = 2.77; CI = 1.18–6.48] and among those who lived in provinces other than Ontario and Quebec [aOR = 1.88; CI = 1.08–3.28]. There were no significant differences in COVID-19 symptoms between males and females, however, we did find a significant association between the province, ethnicity, and reported COVID-19 symptoms for female respondents but not for males. The likelihood of having COVID-19-related symptoms was also lower among respondents whose total income was $100,000 or more in 2019 [aOR = 0.18; CI = 0.07–0.45], and among those aged 45–64 [aOR = 0.63; CI = 0.41–0.98] and 65–84 [aOR = 0.42; CI = 0.28–0.64]. These latter associations were stronger among non-visible minorities. Among visible minorities, being black or of the mixed race and living in Alberta were associated with higher odds of COVID-19-related symptoms.

## Conclusion

We conclude that ethnicity, age, total income in 2019, and province were significantly associated with experiencing COVID-19 symptoms in Canada. The significance of these determinants varied by gender and minority status. Considering our findings, it will be prudent to have COVID-19 mitigation strategies including screening, testing, and other prevention policies targeted toward the vulnerable populations. These strategies should also be designed to be specific to each gender category and ethnic group, and to account for minority status.

## Introduction

The first case of the novel Coronavirus, also known as SARS-CoV-2 or COVID-19, was reported in Wuhan, Hubei Province in China in December 2019 [1]. In a few days, on January 13, 2020, Thailand, reported the first case of COVID-19 [2]. Subsequently, the disease spread to multiple countries, and by January 30, 2020, the World Health Organization (WHO) declared COVID-19 a public health emergency of international concern [3]. Thereafter, the spread of the disease led to the WHO declaring it a global pandemic [4]. Canada was not spared of this pandemic, and since its start, the country has recorded more than 3.5 million cases and more than 37,000 deaths [5].

COVID-19 spreads through human contact or respiratory droplets [2]. According to the WHO, the most common symptoms of COVID-19 are fever, dry cough, and tiredness [3, 7]. Other less common symptoms include aches and pains, headaches, a rash on skin, or discolouration of fingers or toes, loss of smell and taste, sore throat, conjunctivitis, and diarrhoea [6, 7]. These symptoms are expected to be developed within 1–14 days after infection; however, the average incubation time is 5–6 days after infection [6]. In severe cases, it could cause pneumonia, respiratory failure, septic shock, or multiple organ dysfunction or failure [6]. The WHO also identifies difficulty breathing or shortness of breath, chest pain or pressure, and loss of speech or movement as some of the serious symptoms of COVID-19 that require immediate medical attention [7].

In countries such as the United States (U.S), a number of studies demonstrate that racial/ethnic minorities and individuals from segregated areas and low-income backgrounds are more likely to become infected and die from COVID-19 [8–12]. Moreover, other important

socio-demographic and biological characteristics including age, gender, and presence of underlying chronic non-communicable problems (e.g., cardiovascular diseases, hypertension, diabetes, obesity, and chronic obstructive pulmonary disease) increase people's risk of COVID-19 infection and death [13, 14].

Concerning Canada, studies assessing the socio-economic determinants of COVID outcomes, and how they vary by identity factors such as gender and ethnic minority status, remain scarce. This is likely due to limited individual-level data on socio-economic characteristics and their relationship with COVID outcomes, which has prevented a systematic assessment of the socioeconomic and demographic determinants of COVID-19 outcomes in this country. Nevertheless, a few studies exist. For instance, Lapointe-Shaw et al. conducted a syndromic analysis of COVID-19 symptoms in Canada, which found that the prevalence of reporting a combination of fever with cough or shortness of breath is higher among visible minorities [15]. Relatedly, St-Denis' study revealed that being a female, older age, and having lower income were associated with a greater risk of exposure [16]; however, their scope was limited to socio-demographic determinants of occupational risk of exposure to COVID-19. Another study found that being a female and having lower education exacerbated infection risk inequities [17]. Similarly, Wu et al.'s study of the level and predictors of COVID-19 symptoms among the Canadian population revealed that 8% of Canadians reported that they and/or one or more household members experienced COVID-19 symptoms and that the risk of COVID-19 symptoms was higher among younger adults and visible minorities [18]. Our study further builds upon the findings of Wu et al. [18], with the difference in the time periods of the two studies, as well as looking more closely at gender and ethnic disparities in the prevalence of COVID-19 symptoms. We analyse how the socioeconomic and demographic determinants of COVID-19 symptoms differ by gender and visible minority status in Canada.

Identity factors including race and gender both have been found to be significant determinants of COVID-19 outcomes, especially in the U.S. A higher prevalence of non-communicable diseases such as diabetes, hypertension, and cardiovascular diseases and low socio-economic status are some of the factors that are found to be associated with a higher prevalence of COVID-19 in the Afro-American population in the U.S. [19–21]. Similarly, COVID-19 infection and related mortality have gender differences as well. Earlier studies have found higher infection among males than females [22, 23], however, there are other studies which have reported females to be more susceptible than males [24, 25]. There are several genetic, hormonal, and epigenetic factors that influence the difference in infection rates between sexes [26]. More clinical and population-level data from large-scale studies in other countries are needed to make informed conclusions about the gender and ethnicity differences in COVID-19 symptoms, infections, and mortality.

By analysing the socioeconomic and demographic determinants of COVID-19 symptoms and examining how these determinants vary by gender identity and visible minority status in Canada, findings from our study would add to the current state of knowledge of the epidemiology of this virus. COVID-19 policies have significantly varied over time and across regions within Canada; because these policies are likely to affect individuals and population subgroups differently [27, 28], the determinants of COVID-19 symptoms can be expected to vary across gender and ethnic groups. Our study is necessary because it is likely to illuminate the demographic nuances peculiar to the Canadian population during pandemics. Also, as new strains of the COVID-19 emerge, it is important to understand the demographic disparities to be able to implement policies and interventions that target and prioritise the most at-risk subpopulations.

## Materials and methods

### Study design and data collection

Using a cross-sectional study design, we collected data via an online SurveyMonkey platform with portals on the University of Ottawa website available in French and English (https://socialsciences.uottawa.ca/research/covid19-survey). The survey was based on the COVID-19 Symptoms & Social Distancing Web Survey designed by Canning et al. [29]. The study involves three waves of data collection. However, for this paper, we have used data from the first wave that was collected between July and October 2020. The survey questionnaire collected demographic information, recent work experiences, loss of income, experiencing symptoms of COVID-19, mental health conditions, and social distancing behaviour. For data collection, we shared the survey on various social networks and encouraged the snowball method. We also shared the University of Ottawa's Web links via sponsored posts on Facebook in English and French via the University's institutional account. Respondents were sampled online using convenience sampling and snowballing. The questionnaire was available in both English and French. The median time respondents spent completing the survey was around 8 minutes for the French survey and around 7 minutes for the English survey. We collected 3,225 individual responses in English and 1,650 in French for a total of 4,875 across Canada. However, for this paper, only respondents who had complete information on reported COVID-19 symptoms and that of members of their households were considered. Hence, a sample size of 2,829 individual respondents was considered for this study.

### Outcome variables

Two outcome variables were considered for the analysis. The first outcome variable was COVID-19 symptoms among respondents. To derive this outcome variable, we used questions in which respondents were asked if they had experienced any of the following symptoms in the past 2 weeks: fever, dry cough, shortness of breath, decreased sense of smell/taste, and other flu-like symptoms. The responses were "yes", "no" and "don't know". Respondents who mentioned that they had experienced at least one of these symptoms were regarded as those experiencing COVID-19 symptoms, and the remaining were categorized as experiencing no COVID-19 symptoms. The second outcome variable was COVID-19 symptoms among either respondent or a household member. This variable was derived from two questions: 1. Has anyone else in your household besides yourself experienced any of the following symptoms in the past 2 weeks? 2. Have you experienced any of the following symptoms in the past 2 weeks? The responses were "yes", "no" and "don't know". The symptoms were those listed above. COVID-19 symptoms among respondents or household members were obtained from affirmative responses provided by respondents of their experience or the experience of a household member of at least one of the symptoms and otherwise were categorized as no COVID-19 symptoms among respondents or household members.

### Exposure variables

Eight variables were considered as exposures in this study. These were gender (female and male), age (18–44, 45–64, 65–84, and 85 years and above), province (Quebec, Ontario, British Columbia, and other), race (white, black, mixed race, other-aboriginal/indigenous, Asian, Latin American, Arab, and other race), belongs to a visible minority group (no and yes), highest level of education, total personal income in 2019, and number of household members. The highest level of education was coded as high school or less, college, and postgraduate. Total income in 2019 was coded as less than $20,000, $20,000 to less than $50,000, $50,000 to less

than $100,000, and $100,000 or more. The number of household members was divided into three categories—1, 2–3, and 4 or more. The choice of these variables was based on their associations with reported COVID-19 symptoms in previous studies [18, 29].

## Statistical analysis

Stata version 14 was used to clean the data, recode variables, and analyse the data. Both descriptive (frequencies and percentages) and inferential (chi-square test of independence and multivariate logistic regression) analyses were carried out. Frequencies and percentages were first used to describe the socio-demographic characteristics of the respondents and present the proportions of reported COVID-19 symptoms among the respondents and their household members. Next, the chi-square test of independence was used to show the difference in reported COVID-19 symptoms among the respondents and their household members across the socio-demographic characteristics of the respondents. Statistical significance was obtained at a 95% confidence interval. Finally, multivariable logistic regression was employed to examine the socio-demographic characteristics associated with reported COVID-19 symptoms among the respondents only and with symptoms reported by respondents or a household member. We estimated four different models. Model 1 had gender and age of respondents as exposure variables. In Model 2, province, race, and visible minority group were added to the variables in Model 1 as controls. In Model 3, the highest level of education and total income in 2019 were added to the variables in Model 2 as controls. In the final model (Model 4), all the socioeconomic and demographic characteristics were included. To assess whether the socio-economic and demographic determinants of the likelihood of self-reporting COVID-19 symptoms differ by gender and visible minority status, we implemented our most conservative multivariable regression analysis (Model 4) separately by gender and visible minority status. The results were presented as adjusted odds ratios (aOR) at 95% confidence interval. Model fitness was checked using Pseudo R-squared goodness of fit test, with the highest value indicating the best fit model. Since the survey sample was not representative of the general population, both descriptive statistics and regression results used weights that were generated using the distributions of gender, age, and province from the Demographic Estimations program at StatCan to have national representativeness and correct imbalances between the survey sample and the population. We computed a sampling weight for each sex-age-province group. Specifically, for each sex-age-province group, the weight is equivalent to the ratio of the share (probability of selection) of this group in the 2016 census and the share of the same group in the survey sample. We followed the Strengthening the Reporting of Observational Studies in Epidemiology (STROBE) reporting guidelines to report our results.

## Ethical considerations

Ethics approval was obtained from the Office of Research Ethics and Integrity of the University of Ottawa (S-06-20-5911). The survey was open only to residents of Canada over the age of 18. Respondents had to give signed written consent forms after reading the introduction script. The sensitive questions could be skipped by the participants. The analytical dataset did not include any personal identifiers.

## Results

### Socioeconomic and demographic characteristics of respondents

This section of the results provides information on the socioeconomic and demographic characteristics of respondents. Of the 2,829 respondents in this study, 51.05% were females, and

**Table 1. Descriptive statistics on socio-demographic characteristics of respondents.**

| Variables | Unweighted Frequency | Weighted Frequency | Weighted Percentage |
|---|---|---|---|
| **Gender** | | | |
| Female | 2225 | 1444 | 51.05 |
| Male | 604 | 1384 | 48.95 |
| **Age group** | | | |
| 18–44 | 1145 | 1277 | 45.15 |
| 45–64 | 865 | 941 | 33.25 |
| 65–84 | 807 | 555 | 19.61 |
| 85 years and above | 12 | 56 | 1.99 |
| **Province** | | | |
| Alberta | 1280 | 319 | 11.29 |
| British Columbia | 935 | 392 | 13.85 |
| Ontario | 959 | 1098 | 38.83 |
| Quebec | 145 | 640 | 22.61 |
| Other | 210 | 379 | 13.41 |
| **Race** | | | |
| White | 2529 | 2418 | 85.49 |
| Black | 79 | 90 | 3.18 |
| Mixed race | 73 | 133 | 4.70 |
| Other | 148 | 187 | 6.62 |
| **Minority group** | | | |
| No | 91.02 | 2478 | 87.58 |
| Yes | 254 | 351 | 12.42 |
| **Highest level of education** | | | |
| High school or less | 355 | 391 | 13.83 |
| College | 1787 | 1838 | 64.97 |
| Postgraduate | 687 | 600 | 21.20 |
| **Total income in 2019** | | | |
| Less than $20,000 | 630 | 666 | 23.55 |
| $20,000 to less than $50,000 | 1015 | 878 | 31.03 |
| $50,000 to less than $100,000 | 969 | 1002 | 35.42 |
| $100,000 or more | 215 | 283 | 10.00 |
| **Number of household members** | | | |
| 1 | 574 | 547 | 19.33 |
| 2–3 | 1657 | 1639 | 57.94 |
| 4 or more | 598 | 643 | 22.73 |
| **Total sample size** | | **2829** | **100.00** |

NB: Race: Other includes Manitoba, New Brunswick, Newfoundland and Labrador, Northwest territories, Nova scotia, Nunavut, Prince Edward Island, Saskatchewan and Yukon

45.15% were aged 18–44 (Table 1). About 38.83% of the respondents were from the Ontario Province, and the majority of the respondents (85.49%) classified themselves as whites. More than one in ten respondents belonged to a minority group. With respect to educational level, most of the respondents had a college education (64.97%), followed by those with postgraduate education (21.20%). About 35% alluded to receiving a total income of between $50,000 to less than $100,000 in 2019. Concerning the number of household members, the majority (58%) of the respondents indicated that they belonged to a 2–3-member household, followed by those in four or more members' households (22.73%).

### Prevalence of reported COVID-19 symptoms

This section provides information on the prevalence of reported COVID-19 symptoms during the period in which the data were collected. Of the 2,829 respondents included in the analysis, 13.38% had experienced at least one COVID-19 symptom. Dry cough was the most common symptom experienced by the respondents (7.65%) (see Fig 1). Approximately 16.82% of the respondents reported that they or at least a member of their household had COVID-19 symptoms. Relatedly, dry cough was the most common symptom experienced by respondents or by members of their households (7.62%) (see Fig 2).

### Distribution of COVID-19 symptoms across the socioeconomic and demographic characteristics of respondents

Table 2 shows the distribution of COVID-19 symptoms across the socioeconomic and demographic characteristics of the respondents. The findings showed a statistically significant association between age, ethnicity, province of residence, and total income in 2019 and reporting of COVID-19 symptoms among respondents. With respect to the reporting of COVID-19 symptoms among respondents or a household member, the findings showed statistically significant associations with age, ethnicity, province, highest level of education, total income in 2019, and number of household members.

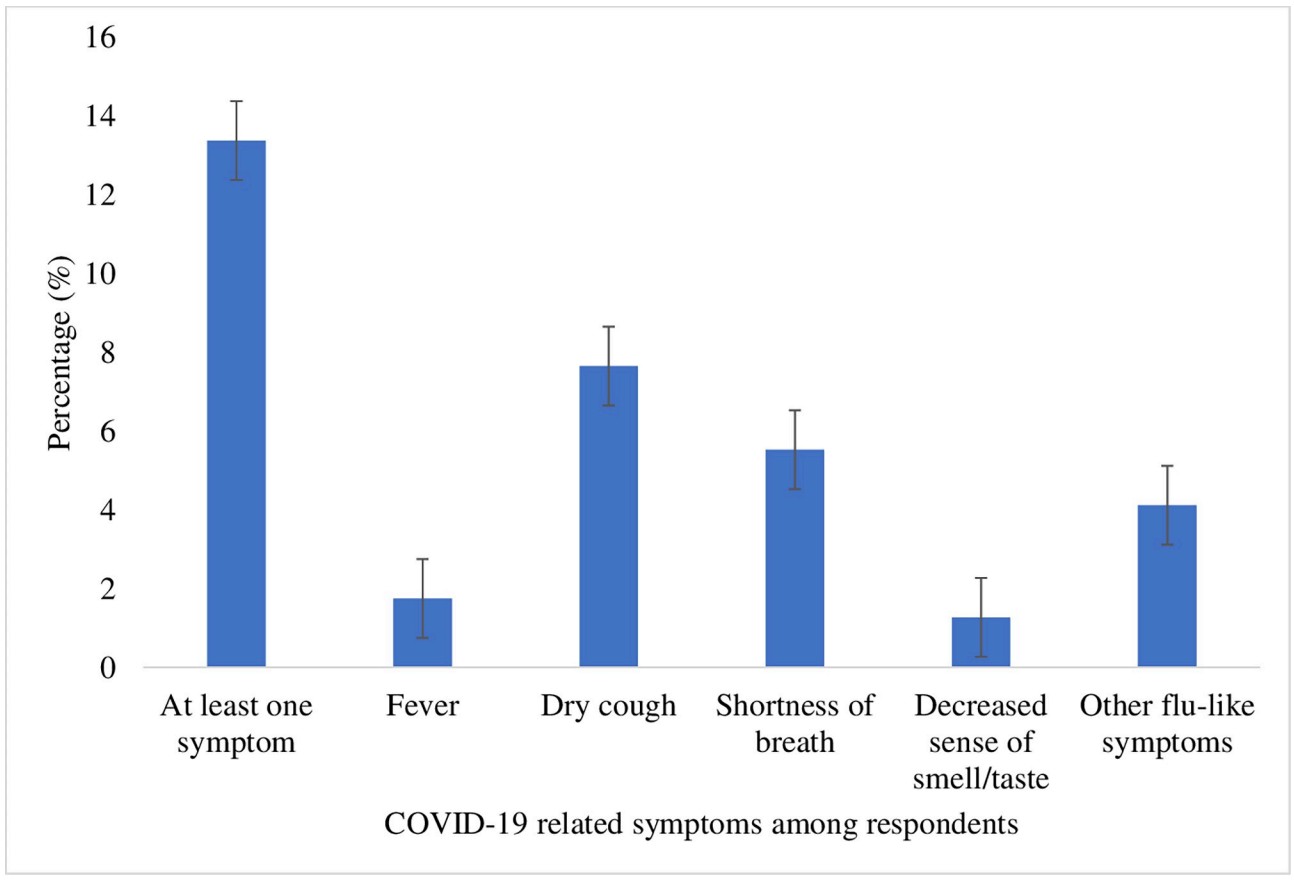

**Fig 1. Prevalence of self-reported COVID-19 symptoms among respondents.**

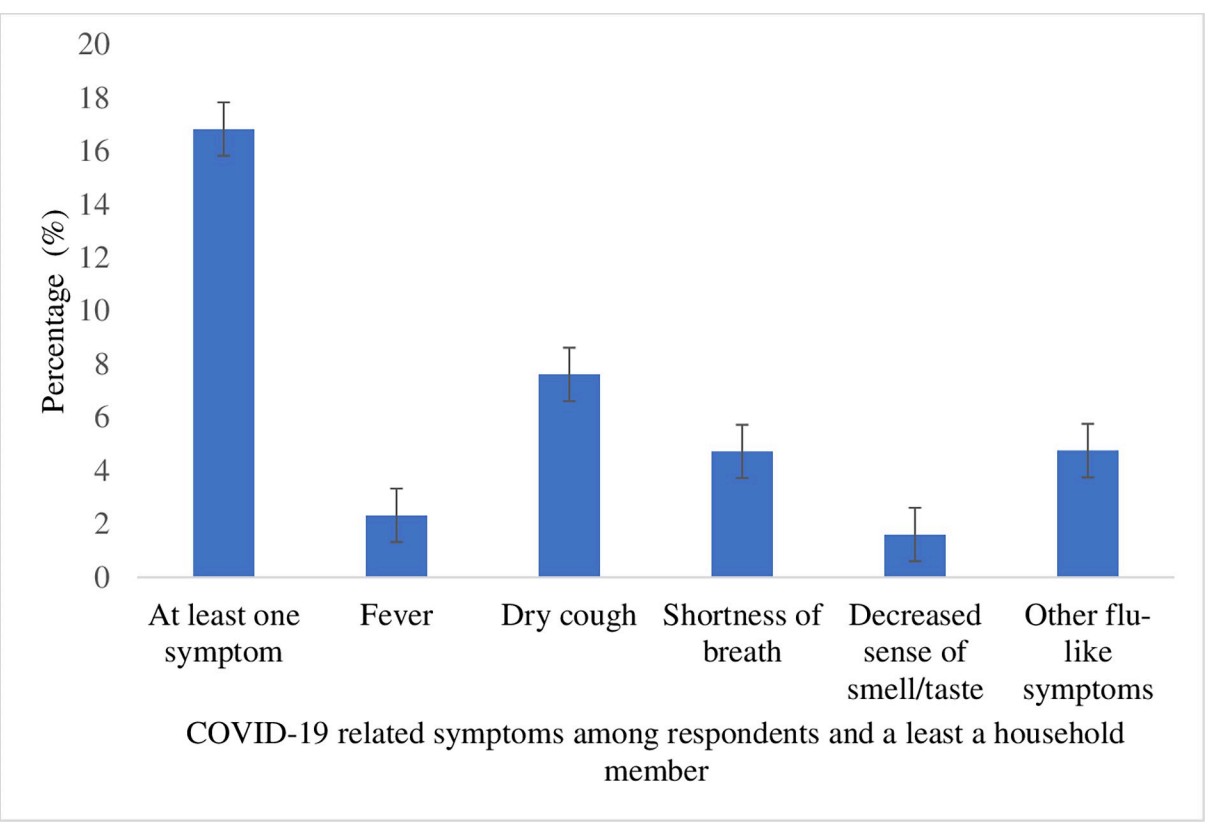

**Fig 2. Prevalence of self-reported COVID-19 symptoms among respondents and a least a household member.**

### Socioeconomic and demographic determinants of COVID-19 symptoms among respondents

This section reports the results of the binary logistic regressions; it highlights the socioeconomic and demographic characteristics of respondents who were more likely to report COVID-19 symptoms. Table 3 presents the results of the association between socioeconomic and demographic characteristics and COVID-19 symptoms among respondents in the past 2 weeks. The analysis revealed that respondents aged 45–64 [aOR = 0.63; CI = 0.41–0.98] and 65–84 [aOR = 0.42; CI = 0.28–0.64] had lower odds of reporting COVID-19 symptoms compared to those aged 18–44. Respondents who lived in "other" provinces (than Alberta, British Columbia, and Ontario) [aOR = 1.88; CI = 1.08–3.28] were more likely to report COVID-19 symptoms compared to those who lived in Quebec. In the same vein, respondents who belonged to mixed race [aOR = 2.77; CI = 1.18–6.48] were more likely to report COVID-19 symptoms compared to whites. The likelihood of reporting COVID-19 symptoms was lower among respondents whose total income was $100,000 or more in 2019 compared to those whose total income was less than $20,000 in 2019 [aOR = 0.18; CI = 0.07–0.45].

### Socioeconomic and demographic determinants of COVID-19 symptoms among respondents and household members

This section reports the results of the binary logistic regressions that analyse the socioeconomic and demographic factors associated with COVID-19 symptoms among respondents or household members in the past 2 weeks. The results are presented in Table 4. Compared to respondents aged

**Table 2. Distribution of COVID-19 symptoms across socio-demographic factors of respondents.**

| Variables | COVID-19 symptoms among respondents only in the past 2 weeks | p-value | COVID-19 symptoms among respondent or their household members in the past 2 weeks | p-value |
|---|---|---|---|---|
| | Yes (%) | | Yes (%) | |
| **Gender** | | 0.986 | | 0.164 |
| Female | 11.97 | | 16.70 | |
| Male | 14.85 | | 16.96 | |
| **Age** | | <0.001 | | <0.001 |
| 18–44 | 18.09 | | 23.06 | |
| 45–64 | 10.33 | | 12.81 | |
| 65–84 | 9.07 | | 9.79 | |
| 85 years and above | 0 | | 11.69 | |
| **Province** | | 0.074 | | 0.003 |
| Alberta | 16.32 | | 20.27 | |
| British Columbia | 7.62 | | 10.34 | |
| Ontario | 12.23 | | 15.28 | |
| Quebec | 12.46 | | 16.34 | |
| Other | 21.75 | | 25.91 | |
| **Race** | | 0.046 | | 0.036 |
| White | 13.03 | | 16.32 | |
| Black | 15.68 | | 18.92 | |
| Mixed race | 25.11 | | 30.93 | |
| Other | 8.50 | | 12.29 | |
| **Minority group** | | 0.309 | | 0.610 |
| No | 13.56 | | 17.04 | |
| Yes | 12.14 | | 15.33 | |
| **Highest level of education** | | 0.156 | | 0.020 |
| High school or less | 14.82 | | 16.03 | |
| College | 12.04 | | 15.57 | |
| Postgraduate | 16.55 | | 21.19 | |
| **Total income in 2019** | | 0.001 | | 0.006 |
| Less than $20,000 | 15.01 | | 17.67 | |
| $20,000 to less than $50,000 | 16.47 | | 19.68 | |
| $50,000 to less than $100,000 | 12.32 | | 16.63 | |
| $100,000 or more | 3.71 | | 6.66 | |
| **Number of household members** | | 0.304 | | <0.001 |
| 1 | 12.45 | | 13.35 | |
| 2–3 | 14.38 | | 17.43 | |
| 4 or more | 11.62 | | 18.23 | |

*P-values obtained from chi-square test

18–44, those aged 45–64 [aOR = 0.57; CI = 0.38–0.85] and 65–84 [aOR = 0.35; CI = 0.24–0.53] were less likely to report that they or at least a member of their households experienced COVID-19 symptoms. In addition, respondents with an income of at least $100,000 were less likely to report that they or at least a member of their households experienced COVID-19 symptoms compared to those with income less than $20,000 [aOR = 30; CI = 0.13–0.67].

**Table 3. Binary logistic regression results on the socio-demographic factors associated with COVID-19 symptoms.**

| Variables | Respondents only having COVID-19 symptoms in the past 2 weeks | | | |
|---|---|---|---|---|
| | aOR [95%CI]; Model 1 | aOR [95%CI]; Model 2 | aOR [95%CI]; Model 3 | aOR [95%CI]; Model 4 |
| **Gender** | | | | |
| Female | Reference | Reference | Reference | Reference |
| Male | 1.22[0.85–1.76] | 1.22[0.85–1.75] | 1.35[0.93–1.94] | 1.30[0.91–1.86] |
| **Age** | | | | |
| 18–44 | Reference | Reference | Reference | Reference |
| 45–64 | 0.52**[0.34–0.80] | 0.51**[0.33–0.79] | 0.65[0.42–1.01] | 0.63*[0.41–0.98] |
| 65–84 | 0.45***[0.31–0.67] | 0.45***[0.30–0.67] | 0.45***[0.30–0.68] | 0.42***[0.28–0.64] |
| 85 years and above | - | - | - | - |
| **Province** | | | | |
| Quebec | | Reference | Reference | Reference |
| British Columbia | | 0.57[0.31–1.04] | 0.62[0.33–1.14] | 0.61[0.33–1.14] |
| Ontario | | 0.97[0.68–1.40] | 1.07[0.73–1.55] | 1.08[0.74–1.57] |
| Alberta | | 1.28[0.63–2.58] | 1.69[0.83–3.44] | 1.67[0.83–3.40] |
| Other | | 1.92*[1.09–3.35] | 1.88*[1.07–3.29] | 1.88*[1.08–3.28] |
| **Race** | | | | |
| White | | Reference | Reference | Reference |
| Black | | 1.42[0.41–4.94] | 1.68[0.50–5.68] | 1.82[0.55–6.03] |
| Mixed race | | 2.18[0.85–5.61] | 2.63*[1.14–6.10] | 2.77*[1.18–6.48] |
| Other | | 0.67[0.22–2.09] | 0.71[0.25–2.07] | 0.75[0.27–2.13] |
| **Minority group** | | | | |
| No | | Reference | Reference | Reference |
| Yes | | 0.63[0.27–1.51] | 0.51[0.23–1.15] | 0.52[0.23–1.16] |
| **Highest level of education** | | | | |
| High school or less | | | Reference | Reference |
| College | | | 1.00[0.56–1.79] | 0.96[0.54–1.71] |
| Postgraduate | | | 1.53[0.78–3.01] | 1.45[0.73–2.86] |
| **Total income in 2019** | | | | |
| Less than $20,000 | | | Reference | Reference |
| $20,000 to less than $50,000 | | | 1.16[0.71–1.89] | 1.13[0.69–1.85] |
| $50,000 to less than $100,000 | | | 0.75[0.45–1.26] | 0.75[0.45–1.25] |
| $100,000 or more | | | 0.12***[0.07–0.43] | 0.18***[0.07–0.45] |
| **Number of household members** | | | | |
| 1 | | | | Reference |
| 2–3 | | | | 1.03[0.66–1.60] |
| 4 or more | | | | 0.74[0.43–1.27] |
| Pseudo R-squared | 0.020 | 0.045 | 0.069 | 0.071 |

Exponentiated coefficients; 95% confidence intervals in brackets

aOR adjusted odds ratios

CI Confidence Interval

* $p < 0.05$

** $p < 0.01$

*** $p < 0.001$

**Table 4. Binary logistic regression results on the socio-demographic factors associated with COVID-19 symptoms.**

| Variables | Respondents or a household member having COVID-19 rsymptoms in the past 2 weeks | | | |
|---|---|---|---|---|
| | AOR [95%CI] Model 1 | AOR [95%CI] Model 2 | AOR [95%CI] Model 3 | AOR [95%CI] Model 4 |
| **Gender** | | | | |
| Female | Reference | Reference | Reference | Reference |
| Male | 0.99 [0.70–1.39] | 0.98[0.70–1.37] | 1.06[0.75–1.49] | 1.07 [0.76–1.49] |
| **Age** | | | | |
| 18–44 | Reference | Reference | Reference | Reference |
| 45–64 | 0.49***[0.34–0.71] | 0.48***[0.32–0.70] | 0.57**[0.38–0.84] | 0.57**[0.38–0.85] |
| 65–84 | 0.36***[0.25–0.51] | 0.36***[0.25–0.52] | 0.35***[0.24–0.51] | 0.35***[0.24–0.53] |
| 85 years and above | 0.44 [0.05–3.57] | 0.36 [0.05–2.52] | 0.31[0.05–1.94] | 0.33 [0.05–2.14] |
| **Province** | | | | |
| Quebec | | Reference | Reference | Reference |
| British Columbia | | 0.57*[0.33–0.99] | 0.63[0.36–1.09] | 0.62[0.36–1.08] |
| Ontario | | 0.90 [0.65–1.25] | 0.97[0.69–1.37] | 0.96[0.69–1.36] |
| Alberta | | 1.20 [0.64–2.23] | 1.47[0.78–2.77] | 1.47 [0.79–2.74] |
| Other | | 1.75* [1.04–2.94] | 1.72*[1.01–2.92] | 1.68 [0.99–2.84] |
| **Ethnic groups** | | | | |
| White | | Reference | Reference | Reference |
| Black | | 1.46[0.48–4.43] | 1.68 [0.57–4.96] | 1.70 [0.58–4.95] |
| Mixed race | | 2.26[0.88–5.77] | 2.65*[0.1.13–6.26] | 2.19 [1.12–6.19] |
| Other | | 0.82[0.33–2.06] | 0.90[0.37–2.17] | 0.90[0.37–2.19] |
| **Minority group** | | | | |
| No | | Reference | Reference | Reference |
| Yes | | 0.60 [0.28–1.30] | 0.51[0.24–1.06] | 0.50 [0.24–1.05] |
| **Highest level of education** | | | | |
| High school or less | | | Reference | Reference |
| College | | | 1.13[0.65–1.94] | 1.13[0.65–1.95] |
| Postgraduate | | | 1.66[0.89–3.10] | 1.68[0.89–3.15] |
| **Total income in 2019** | | | | |
| Less than $20,000 | | | Reference | Reference |
| $20,000 to less than $50,000 | | | 1.23 [0.79–1.92] | 1.22[0.78–1.92] |
| $50,000 to less than $100,000 | | | 0.93 [0.59–1.49] | 0.93[0.58–1.48] |
| $100,000 or more | | | 0.31**[0.14–0.69] | 0.30* [0.13–0.67] |
| **Number of household members** | | | | |
| 1 | | | | Reference |
| 2–3 | | | | 1.23 [0.79–1.92] |
| 4 or more | | | | 1.21 [0.73–1.99] |
| Pseudo R-squared | 0.026 | 0.050 | 0.066 | 0.067 |

Exponentiated coefficients; 95% confidence intervals in brackets

aOR adjusted odds ratios; CI Confidence Interval

* $p < 0.05$

** $p < 0.01$

*** $p < 0.001$

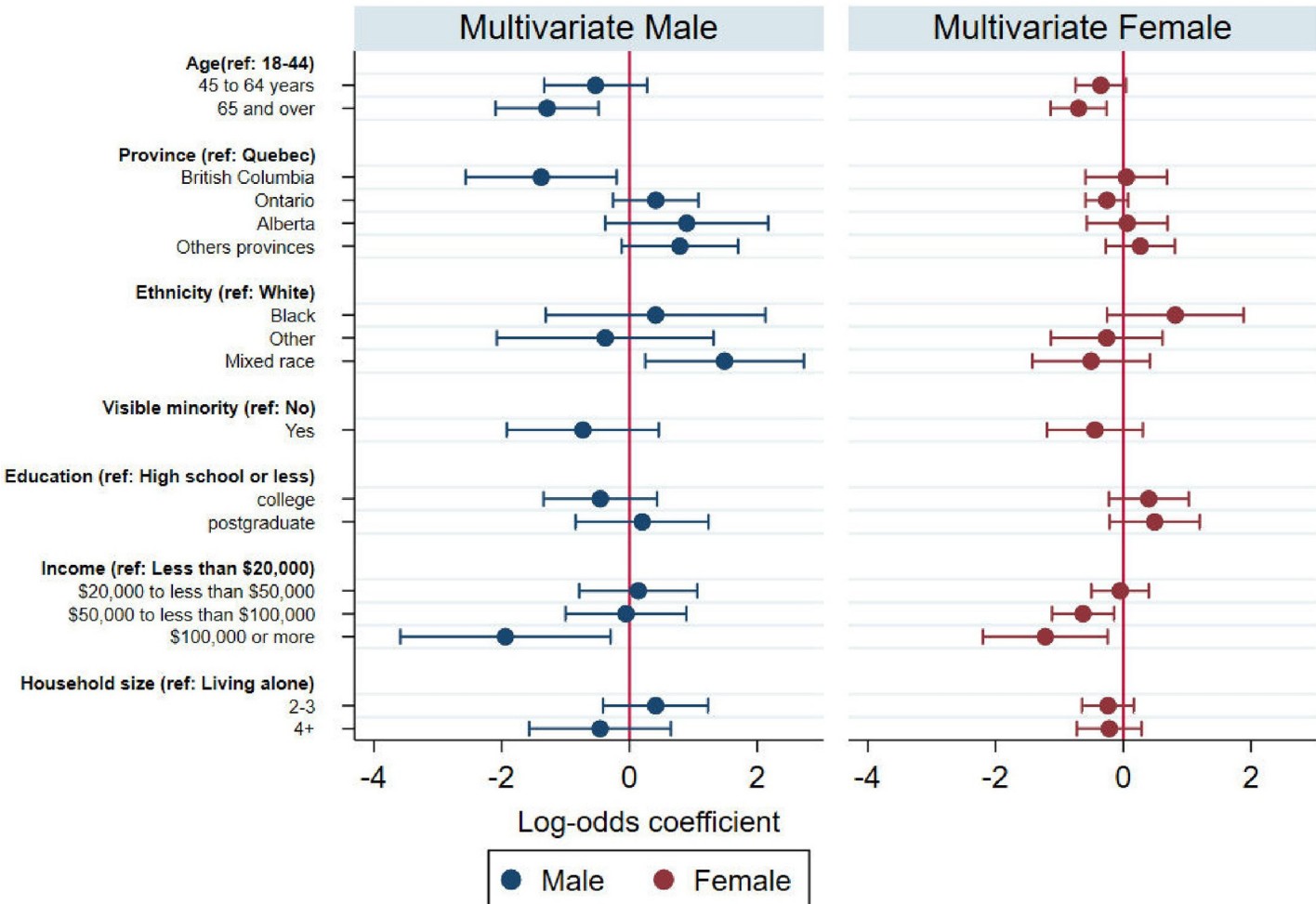

**Fig 3. Socio-demographic factors associated with self-reported COVID-19 symptoms disaggregated by gender.**

### The role of gender: Socioeconomic and demographic factors associated with COVID-19 symptoms disaggregated by gender identity

Fig 3 presents the multivariate logistic regression results of the association between socioeconomic and demographic characteristics and self-reported COVID-19 symptoms disaggregated by gender identity. The results revealed that after controlling for other factors men from mixed race were significantly more likely to report experiencing COVID-19 symptoms while this was not the case for women from mixed race. We also observed that residents of British Columbia were significantly less likely to report COVID-19 symptoms, but this is only true for men and not for women. Additionally, while being black is not significantly associated with the likelihood of self-reporting COVID-19 for both men and women, the size of the odd ratio suggests that Black men and Black women have higher odds of reporting COVID-19 symptoms, but the effect is more pronounced for Black women.

Fig 4 presents the results of the association between socioeconomic and demographic characteristics and reporting of COVID-19 symptoms among respondents and at least a household member disaggregated by gender. The results revealed that both older men and women were less likely to report that they or at least a member of their households experienced COVID-19

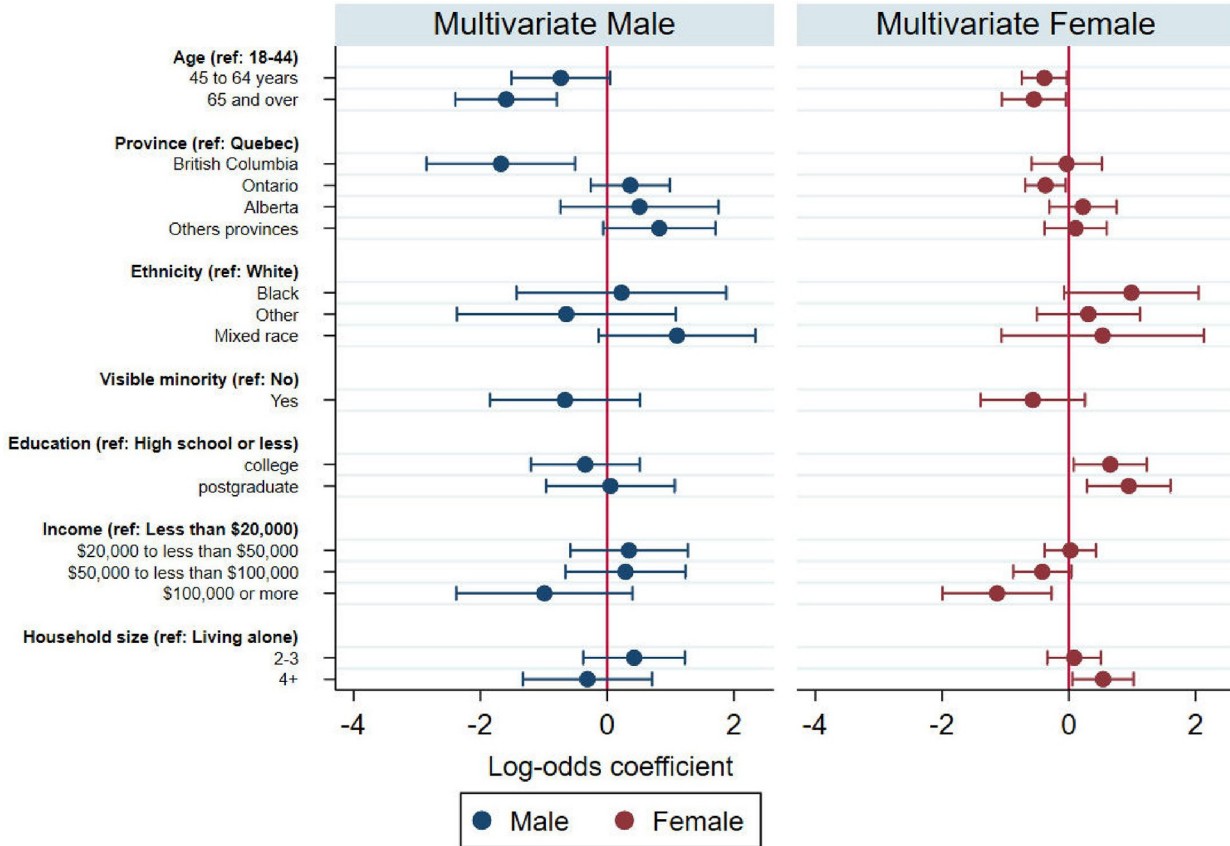

**Fig 4. Socio-demographic factors associated with COVID-19 symptoms among respondents and household members by gender.**

symptoms compared to younger men and women. While men who lived in British Columbia were significantly less likely report that they or at least a member of their households experienced COVID-19 symptoms compared to those who lived in Quebec, females who lived in Ontario were significantly less likely to report that they or at least a member of their households experienced COVID-19 symptoms compared to those who lived in Quebec. Highest level of education and number of people in a household was a significant predictor of experience of COVID-19 symptoms among female respondents, but this was not the case for male respondents.

## The role of minority status: Socioeconomic and demographic factors associated with COVID-19 symptoms disaggregated by visible minority status

This section highlights the socioeconomic and demographic factors associated with self-reported COVID-19 symptoms disaggregated by minority status. Fig 5 reports the multivariate logistic regression examining the socioeconomic and demographic factors associated with self-reported COVID-19 symptoms among respondents. The analysis revealed that in both visible minorities and non-visible minorities, respondents who earned $100,000 or more were less likely to report that they experienced COVID-19 symptoms, but this association tended to be higher among non-visible minorities. In addition, within visible minorities, respondents who were black or mixed race or living in Alberta had higher odds of presenting COVID-19 symptoms. By comparison, among the non-visible minorities, male respondents and those living in

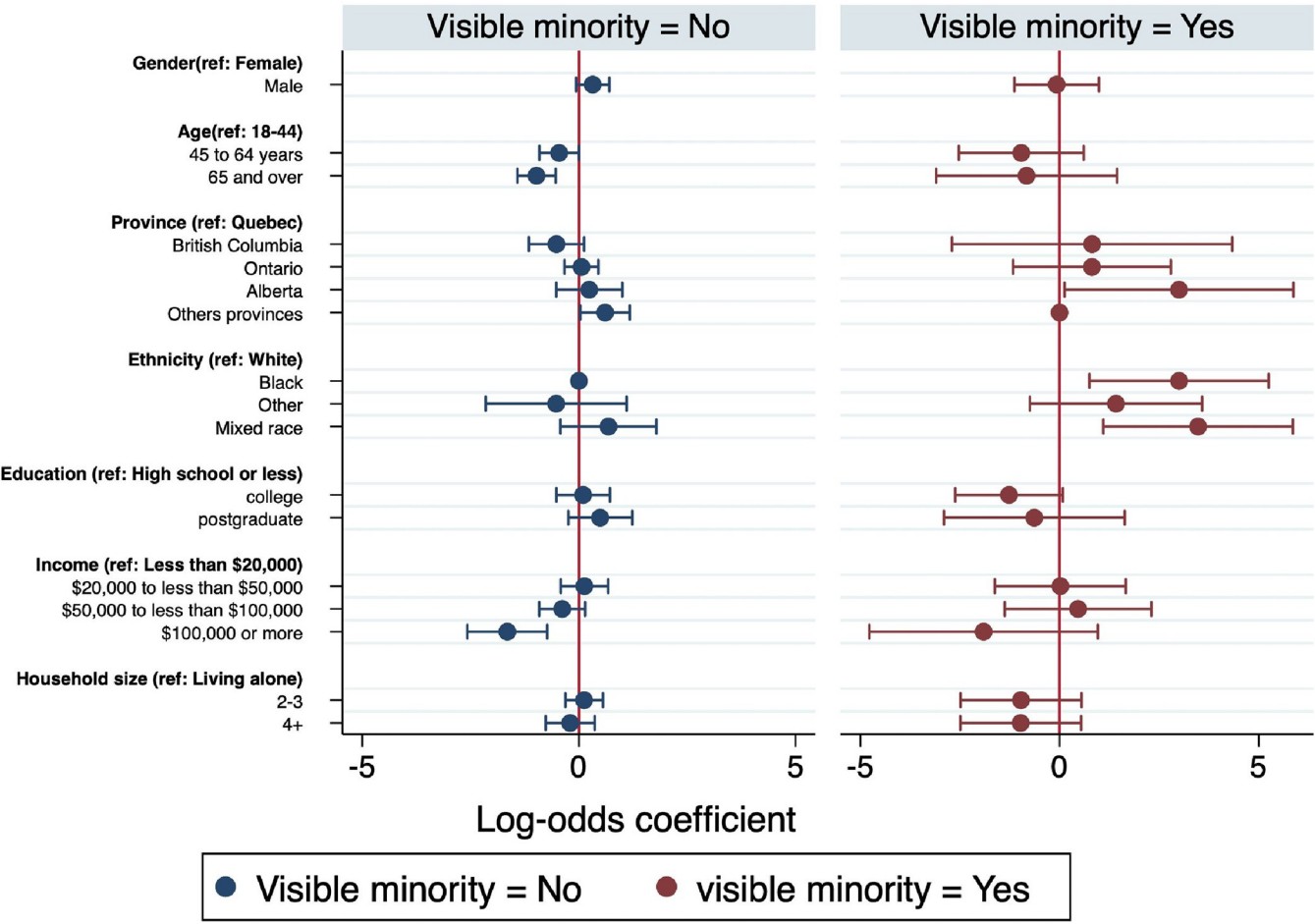

**Fig 5. Socio-demographic factors associated with COVID-19 symptoms among respondents and household members by visible minority.**

provinces other than Alberta, Ontario, and British Columbia were more likely to report they experienced COVID-19 symptoms, while those who were older were less likely to present COVID-19 symptoms.

Fig 6 presents the socioeconomic and demographic factors associated with self-reported COVID-19 symptoms among respondents and at least a member of their household disaggregated by visible minority status. The analysis revealed that among respondents who were from a visible minority group, those who were older and those who earned $100,000 or more were less likely to report that they or at least a member of their households experienced COVID-19 symptoms. These associations were less strong among non-visible minorities. In addition, among respondents who were of visible minority, those who lived in Alberta and those of the black, mixed, and other race were more likely to report that they or at least a member of their households experienced COVID-19 symptoms.

## Discussions

Our analyses assessed the socioeconomic and demographic factors associated with the reporting of COVID-19 symptoms in Canada, and how these determinants varied by gender and

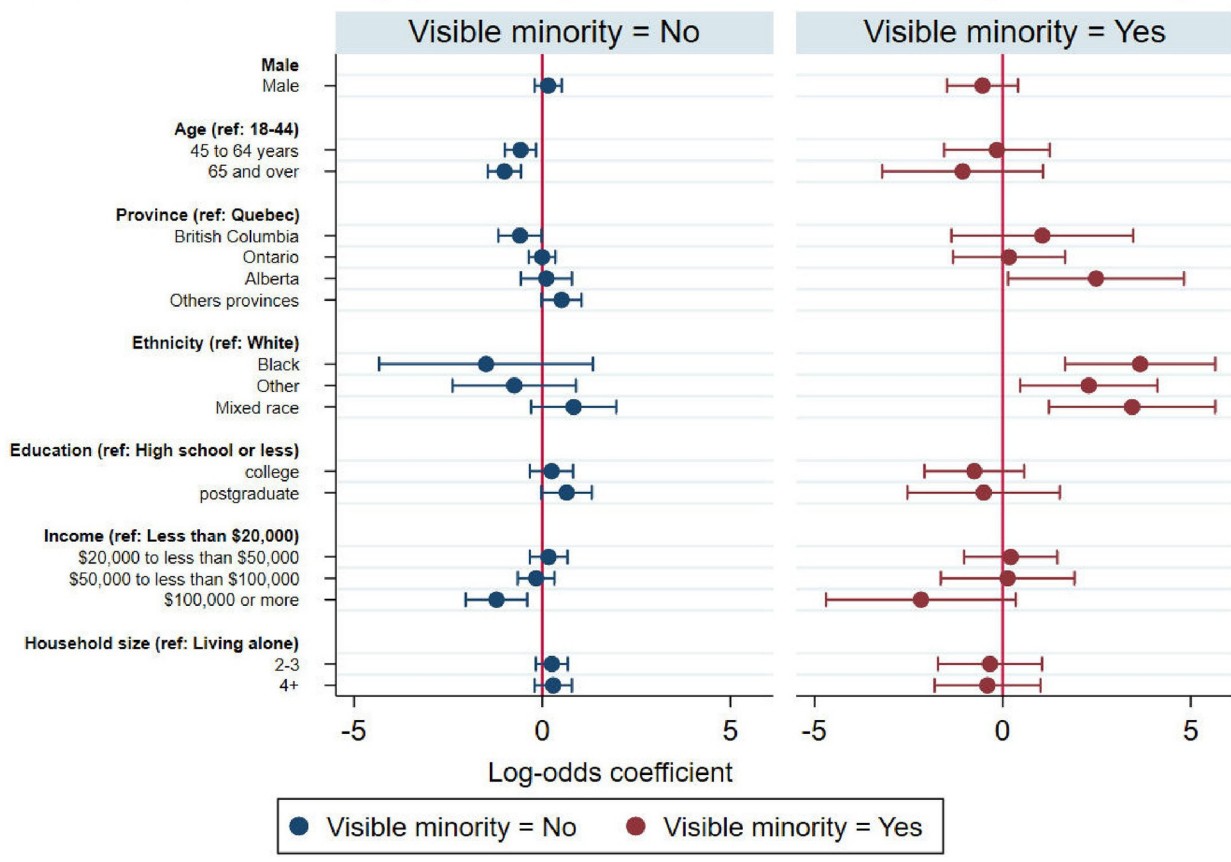

**Fig 6. Socio-demographic factors associated with COVID-19 symptoms among respondents and at least a member of the household disaggregated by visible minority.**

visible minority status. Overall, the prevalence of self-reported COVID-19 symptoms was relatively higher at the individual level (12.3%) compared to COVID-19 symptoms self-reported among household members (6.9%). This is higher than the 8% that was reported by Wu et al. [18] and higher than the 1.6% COVID symptoms reported in Lapointe-Shaw et al.'s study [15]. Although Canada has fared relatively well in terms of the number of cases and deaths due to COVID-19 when compared to other developed countries, especially its neighbour, the U.S., the distribution of the cases and deaths within Canada was not homogenous. There were interprovincial differences in self-reported COVID-19 symptoms in our study. Respondents residing in "other" provinces (than Alberta, British Columbia, and Ontario) had greater likelihood of reporting COVID-19 symptoms compared to those residing in Quebec. Also, as with other health, social, and economic issues, COVID-19 disproportionately affected the vulnerable populations in Canada [30]. There are a number of studies elucidating the inequities in COVID-19 cases and deaths, and the disproportionate burden born by the marginalized population in countries most affected by the COVID-19 pandemic such as the U.S., the UK, Italy, and France [31–34].

The prevalence of self-reported COVID-19 symptoms was higher among the younger age groups. The age distribution of COVID-19 symptoms is in line with related studies from the United States and Brazil [35, 36]. As noted in other countries, the severity of the infection was quite low in younger age groups [37, 38]. Compared to respondents aged 18–44, reporting of COVID-19 symptoms was lower among respondents aged 45–84 and lower for at least a

member of their households. Our findings corroborate those of Wu et al. who found the reported symptoms of COVID-19 lower among older adults than any other age group [18]. Plausibly, it could be a limitation of our study (probably sampling bias) in which only those older adults who participated in the survey were healthier, living in a residence, and had access to the internet. Consequently, the observed lower odds of reporting COVID-19 symptoms among older adults (i.e., age 65+) is reflective of a significant limitation of our study; that is, the lack of representation of persons in nursing and long-term care homes that are inhabiting majority of the Canadian older people, and where COVID-19 cases and deaths have been the highest [39]. As contended by Graham [40], older adults often report symptoms such as dizziness and confusion which are also symptoms of COVID-19; however, our study did not consider dizziness and confusion as COVID symptoms. Another explanation for lower reporting of COVID-19 symptoms among older adults could be because of their health behaviour, which is usually preventive and cautious [41].

In contrast to the findings of a syndromic analysis that found no significant difference in reported symptoms of COVID-19 across income groups in Canada [15], we found significant association between income levels of individuals and their reporting of COVID-19 symptoms. Per our study, the risk of reporting symptoms reduced significantly with increasing income. This finding corroborates a similar study by Mena et al. that showed a strong association between socioeconomic status and the risk of COVID-19 symptoms, incidence, and mortality in Chile [42]. People belonging to lower-income groups are usually employed in low-paying essential jobs, where working from home is not an option, and thus are at a higher risk of getting infected [43, 44].

Consistent with previous studies [45, 46], we found larger household size to be significantly associated with higher odds of reporting COVID-19 symptoms among household members. This finding reflects the compounded risk emanating from having more household inhabitants. Large household size makes it difficult to observe social distancing protocols and reduces the possibility of self-isolation when a household member gets sick or is infected [47, 48].

Males and females had an equal prevalence of experiencing COVID-19 symptoms. Related studies have found that, while men and women show comparable odds of experiencing COVID-19 symptoms [49, 50], men with COVID-19 are more at risk for poorer outcomes and death [50]. Besides typical male-female differences [51, 52], processes unique to COVID-19 probably play a role in this mortality disparity [53]. Male-female characteristics and unique COVID-19 processes may also explain the discrepancies in the socioeconomic and demographic factors determining COVID-19 symptoms by gender. The province of residence and ethnicity were significantly associated with self-reported COVID-19 symptoms among female respondents but not among male respondents in our study. When the analysis was disaggregated by gender, we found that males who lived in British Columbia were less likely to self-report COVID-19 symptoms compared to those who lived in Quebec. Among females, province was not a significant factor associated with self-reported COVID-19 symptoms. Evidence shows that Ontario and Quebec, which are the two most populous provinces had reported the highest infection and cases of COVID-19 [49]. Hence, stringent preventive measures were instituted in these two provinces compared to the other provinces.

Our results also revealed that among male respondents, those from mixed race were significantly more likely to report experiencing COVID-19 symptoms while this was not the case for women from mixed race. This racial disparity is corroborated by the findings of a systematic review conducted among adults in the UK and the U.S. [53]. Similarly, our findings are consistent with a related study conducted in the U.S. [54]. There are multiple reasons for the observed racial and sex disparities, most of which the present study does not directly account for. For instance, our findings may be explained by masculinity norms that permeate mixed-

race populations [54]. Masculinity ideals that are upheld by men of mixed race are negatively correlated with their adherence to COVID-19 preventive measures; hence, significantly increasing their risk of reporting more COVID-19 symptoms. Additionally, Black men and Black women have higher odds of reporting COVID-19 symptoms, but the effect is more pronounced for Black women. This result is consistent with previous studies [55–59]. Primarily, the result may be due to structural factors from occupation and access to healthcare [54]. In the U.S., Frye [55] noted that there is an overrepresentation of black women working as nurse assistants and home health aides. Such essential occupations exacerbate black women's risk of reporting COVID-19 symptoms.

Analyses disaggregated by minority status showed that higher income was associated with lower odds of reporting COVID-19 symptoms both among visible minorities and non-visible minorities, but this association tended to be stronger among non-visible minorities. Among visible minorities, respondents of the black or mixed race were more likely to experience COVID-19 symptoms. These findings complement studies that found that racial and ethnic minority groups have disproportionally higher rates of developing COVID-19 illness [60]. Individuals from historically marginalised racial and ethnic groups have been found to suffer disproportionately from frequent and severe medical disorders that increase their risk of experiencing COVID-19 symptoms [61]. We complement this existing literature by documenting disparities in COVID-19 symptoms among visible minority ethnic groups. We also found that age was significantly associated with COVID-19 self-reported symptoms among non-visible minorities. Older respondents were less likely to present COVID-19 symptoms. This could be explained by the fact that Canadian seniors were more concerned about their health and took more precautions than younger individuals [62]. Our analysis however suggests that this was more likely to be the case among non-visible minorities.

Overall the findings indicate that some sub-populations including individuals from low-income earners, older adults, those of the mixed race, and those who lived in "other" provinces (than Alberta, British Columbia, and Ontario) were at high risk of reporting COVID-19 symptoms. The socioeconomic and demographic determinants of COVID-19 symptoms also varied by gender and visible minority status. The analysis implies that it is imperative to strengthen current preventive interventions in vulnerable sub-populations.

## Strengths and limitations

Our study substantiates other studies conducted in Canada that established an association between socioeconomic and demographic factors and COVID-19 symptoms at the individual and household levels. Moreover, the sample used for this study is nationally representative and facilitates the generalisability of the findings to the larger Canadian population. Nevertheless, our findings should be interpreted while taking into consideration some limitations. The study design was cross-sectional and as such, causality cannot be easily established. Also, this study used self-reported data, hence, it is likely that there may have been some recall bias. Additionally, respondents were sampled online using convenience and snowballing, which can induce noise to our results. Moreover, internet-based surveys usually have sampling bias. We tried to overcome this limitation by our sample. Finally, a large share of the older population of Canada is unlikely to complete a survey on a web platform. These include nursing home residents and those with chronic health issues or disabilities.

## Conclusion

Our findings suggest that COVID-19 mitigation strategies including screening, testing, and containment should focus on the vulnerable populations (i.e., low-income earners, older

adults, those of the mixed race, and those who lived in provinces other than Alberta, British Columbia, and Ontario). The socioeconomic and demographic determinants of COVID-19 symptoms vary by identity factors including gender and visible minority status in Canada. The analysis suggests that mitigation strategies should be designed to be specific to each gender category, ethnic group, and minority status in Canada.

## Author Contributions

**Conceptualization:** Roland Pongou, Sanni Yaya.

**Formal analysis:** Roland Pongou, Bright Opoku Ahinkorah, Marie Christelle Mabeu, Sanni Yaya.

**Funding acquisition:** Roland Pongou, Sanni Yaya.

**Investigation:** Stéphanie Maltais.

**Methodology:** Arunika Agarwal, Sanni Yaya.

**Project administration:** Roland Pongou.

**Supervision:** Sanni Yaya.

**Validation:** Stéphanie Maltais, Sanni Yaya.

**Writing – original draft:** Roland Pongou, Bright Opoku Ahinkorah, Sanni Yaya.

**Writing – review & editing:** Roland Pongou, Bright Opoku Ahinkorah, Marie Christelle Mabeu, Arunika Agarwal, Stéphanie Maltais, Aissata Boubacar Moumouni, Sanni Yaya.

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
