## [Decision Letter · Decision Letter 0]

16 Jun 2022

PGPH-D-22-00694

Gender and ethnic disparities in COVID-19 related symptoms in Canada: Evidence from a national cross-sectional survey

Dear Dr. Sanny Yaya,

Thank you for submitting your manuscript to PLOS Global Public Health. After careful consideration, we feel that it has merit but does not fully meet PLOS Global Public Health’s publication criteria as it currently stands. Therefore, we invite you to submit a revised version of the manuscript that addresses the points raised during the review process.

We look forward to receiving your revised manuscript.

Kind regards,

Haroon Ahmed, PhD

Academic Editor

Journal Requirements:

1. Please update the 'Competing Interests' statement,please state "The authors have declared that no competing interests exist".

- State the initials, alongside each funding source, of each author to receive each grant.

- State what role the funders took in the study. If the funders had no role in your study, please state: “The funders had no role in study design, data collection and analysis, decision to publish, or preparation of the manuscript.”

3. Please provide separate figure files in .tif or .eps format.

4. In the online submission form, you indicated that "The datasets generated during and/or analysed during the current study are available from the corresponding author on reasonable request.". All PLOS journals now require all data underlying the findings described in their manuscript to be freely available to other researchers, either 1. In a public repository, 2. Within the manuscript itself, or 3. Uploaded as supplementary information.

Additional Editor Comments (if provided):

Reviewers' comments:

Reviewer's Responses to Questions

**Comments to the Author**

1. Does this manuscript meet PLOS Global Public Health’s publication criteria? Is the manuscript technically sound, and do the data support the conclusions? The manuscript must describe methodologically and ethically rigorous research with conclusions that are appropriately drawn based on the data presented.

Reviewer #1: Partly

Reviewer #2: Partly

Reviewer #3: Yes

Reviewer #4: Yes

2. Has the statistical analysis been performed appropriately and rigorously?

Reviewer #1: No

Reviewer #2: I don't know

Reviewer #3: Yes

Reviewer #4: Yes

3. Have the authors made all data underlying the findings in their manuscript fully available (please refer to the Data Availability Statement at the start of the manuscript PDF file)?

Reviewer #1: No

Reviewer #2: Yes

Reviewer #3: Yes

Reviewer #4: No

4. Is the manuscript presented in an intelligible fashion and written in standard English?

Reviewer #1: Yes

Reviewer #2: Yes

Reviewer #3: Yes

Reviewer #4: Yes

5. Review Comments to the Author

Reviewer #1: This manuscript reports the results of an online survey investigating the relationships between socio-demographic variables and COVID-19 symptoms. In general the manuscript is well written, although there are grammatical errors so I encourage the authors to carefully review the text. There are several areas (outlined below) where, in its current form, I do not believe the manuscript meets the publication criteria set by PLoS Global Public Health.

1. Title. "Gender and ethnic disparities ..." does not seem to reflect the results of the study. None of the statistical models found differences in the odds of reporting COVID-19 symptoms between males and females.

2. The data availability statement indicates data may be available from the corresponding author upon reasonable request. This does not meet the journal requirement of data being fully available at the time of publication.

3. The methods section does not contain sufficient detail about the sampling method or target population. For instance, there is limited information about the survey per se, but nothing related to the ampling methdology such as eligibility criteria, how the survey was advertised to potential participants or the sampling frame. There is a sentence in the 'strengths and limitations' section indicating that convenience and snowballing were used, but this is the only mention of the sampling methodology in the manuscript.

3. There are numerous discrepancies between the information in the methods and results. Three examples are:

a) Outcome variable. From the methods "The second outcome variable was COVID-19 related symptoms among either respondent or a household member." The reported results were that 13% of respondents has at least one COVID-19 symptom, while 7% reported they or at least one household member had symptoms. Since the second outcome includes symptoms in the repondent or any household member I do not understand how the prevalence can be lower than that for the respondents. Therefore I cannot make sense of any results reported for the second outcome.

b) Gener-specific models. The methods indicate that Model 3 was the basis for the gender-specific models. This model contained all variables with the exception of number of household members (according to the results in Tables 3 and 4). However, Figure 3 which describes the results of the gender-specific models includes household size.

c) The age groups mentioned in the methods are not the same as those presented in the results.

4. Data were collected for race and minority group, and both were included in the statistical models. There seems to be potential for these two variables to be related. Have the authors assessed whether these variables show enough independence from each other to both be included in the models (eg, do the 351 "yes" responses for minority group come from specific races, rather than across all races)?

5. Table 4. Several entries in Table 4 are incorrect. Model 3 aOR for >$100,000 is outside the 95% CI; Model 4 aOR for number of household members do not look correct in relation to the 95% CI

6. Conclusion. In the conclusion older individuals and individuals in larger households are used as examples of vulnerable populations that should be the focus of COVID-19 mitigation strategies. However, the results from this study do not support these as vulnerable populations. Older individuals had lower odds of COVID-19 symptoms, while the odds of symptoms at an individual level was not related to household size. It was only when the outcome was symptoms in respondent or >=1 household member that the size of the household become significant which is not surprising since the probability of at least one member having symptoms should increase with the number of people at risk.

The authors may also like to consider the following suggestions:

7. Given the non-specific nature of many COVID-19 symptoms it is interesting that the authors have choosen to use presence of a single symptom, rather than the WHO case definition which requires fever AND cough, or at least 3 symptoms. How sensitive are your results to the definition of COVID-19 symptoms? Would the results change if fever AND cough were required, rather than a single symptom? At the moment the study could relate to any infectious disease having fever or cough as a symptom, rather than COVID-19 specifically.

8. It is not clear to me why the results of Models 1 to 3 need to be presented. It appears the conclusions are drawn from Model 4, and there are no real differences in aOR as additional variables are added to the models.

9. It would be helpful if Table 1 included the actual frequencies, along with the weighted frequencies. It is mentioned that the sample was not representative, but there is no corresponding information about which groups are over-sampled or under-sampled.

10. The income variable is personal income. It is not clear how useful this information is for the household outcome since the respondent may have very different characteristics to other members of the household. In the discussion it is concluded that lower incomes means people are more exposed due to the nature of lower-paying jobs. This is an over-generalisation when you don't know the circumstances of the household. For instance, there is a big difference if the respondent income is the only income for the household versus if the respondent is not the main income-earner in the household.

Reviewer #2: Overall:

The findings are potentially of interest, but given the fact that similar studies have already been published and the results widely reported, additional comparison is needed to provide sufficient justification of these results as a standalone publication.

I would recommend adding a comparison to other sources and previous publications (as referenced in the introduction). Are these results aligned with or in contradiction to these other sources and findings? How do these results compare to other countries? These should be included not only in the tables, but in charts to more clearly communicate the results.

In addition, the authors mention the fact that data were collected for subsequent waves; I would suggest including those data here to expand the time scale and basis for analysis.

Specific comments:

Page 6: Missing word in first sentence, second paragraph.

Materials and Methods: How was the survey disseminated? It is clear that it was a convenience sample, but where was it advertised and how were people encouraged to participate? Please provide additional information about recruitment for the convenience sample.

The charts in the figures need to include units on the y-axis and appear to be lacking both figure titles and legends.

Reviewer #3: This manuscript investigated the association between socio-demographic factors (such as gender, age, province, race, minority groups, and level of education) and COVID-19 related symptoms in Canada. An online survey was conducted to collect the socio-demographic data. This study suggested that mitigation strategies should focus on the vulnerable populations. The structure of this manuscript is clear, and the writing is well. However, this paper needs further improvements.

My comments are:

(1) Many studies have thoroughly investigated the impact of socio-demographic factors and other factors on the spreading of COVID-19 in various areas. They have confirmed that the scale of COVID-19 pandemics is associated with socio-demographic factors. Published papers concluded that vulnerable populations need more attention during the COVID-19 pandemics. The results and conclusions of this manuscript provided limited additional information on the impact of socio-demographic factors on COVID-19 pandemics. The necessity of this manuscript should be highlighted. Please highlight the necessity and innovation in the abstract and background sections.

(2) Materials and Methods: The authors used data from the first wave collected between July and October 2020. However, the infectious capacities of different SARS-CoV-2 variants and the symptom caused by SARS-CoV-2 variants were distinct. I suspect that the results of this paper cannot be used to guide the future prevention and control of the COVID-19 pandemics.

(3) Page 5, Exposure variables: please explain why you chose these eight exposure variables and give the references.

(4) How did you deal with the potential collinearity problem among independent variables?

(5) Results: I recommend the authors state the necessity and the logical role of the subsection at the beginning of each subsection.

(6) Discussion: it is better to discuss how can the results and conclusions of this manuscript illuminate the mitigation ways of the pandemics caused by SARS-CoV-2 variants of concern, such as the Omicron variants of concern.

Reviewer #4: 1. The manuscript is technically sound and well written and meets the publication criteria for PLOS Public Health. Methodology applied to answer the research question was appropriate.

2. Statistical analysis was done appropriately. The authors made sure to explain how they arrived to the final model that was used in logistic regression, to make it easier to follow, perhaps the authors should report only those results from the final statistical model. Since the paper is not about comparing statistical models, this should not be the focus of the results section.

3. The authors mentioned that they will make the data available upon request.

4. The manuscript is well written, save for very minor issues in commas and semi-colons in the introduction section and consistency of using abbreviations versus full name (e.g. WHO and World Health Organization)

6. PLOS authors have the option to publish the peer review history of their article (what does this mean?). If published, this will include your full peer review and any attached files.

**Do you want your identity to be public for this peer review?** For information about this choice, including consent withdrawal, please see our Privacy Policy.

Reviewer #1: No

Reviewer #2: No

Reviewer #3: No

Reviewer #4: No

---

## [Decision Letter · Decision Letter 1]

11 Nov 2022

PGPH-D-22-00694R1

Gender and ethnic disparities in COVID-19 related symptoms in Canada: Evidence from a national cross-sectional survey

Dear Dr. Yaya S,

Thank you for submitting your manuscript to PLOS Global Public Health. After careful consideration, we feel that it has merit but does not fully meet PLOS Global Public Health’s publication criteria as it currently stands. Therefore, we invite you to submit a revised version of the manuscript that addresses the points raised during the review process.

EDITOR: The manuscript can be accepted for the publication after the incorporation of minor comments of Reviewer 4.

Please submit your revised manuscript by 20-11-2022. If you will need more time than this to complete your revisions, please reply to this message or contact the journal office at globalpubhealth@plos.org. Please include the following items when submitting your revised manuscript:

We look forward to receiving your revised manuscript.

Kind regards,

Haroon Ahmed, PhD

Academic Editor

Journal Requirements:

Additional Editor Comments (if provided):

Reviewers' comments:

Reviewer's Responses to Questions

**Comments to the Author**

1. If the authors have adequately addressed your comments raised in a previous round of review and you feel that this manuscript is now acceptable for publication, you may indicate that here to bypass the “Comments to the Author” section, enter your conflict of interest statement in the “Confidential to Editor” section, and submit your "Accept" recommendation.

Reviewer #2: All comments have been addressed

Reviewer #3: All comments have been addressed

Reviewer #4: (No Response)

2. Does this manuscript meet PLOS Global Public Health’s publication criteria? Is the manuscript technically sound, and do the data support the conclusions? The manuscript must describe methodologically and ethically rigorous research with conclusions that are appropriately drawn based on the data presented.

Reviewer #2: Yes

Reviewer #3: Yes

Reviewer #4: Yes

3. Has the statistical analysis been performed appropriately and rigorously?

Reviewer #2: Yes

Reviewer #3: Yes

Reviewer #4: Yes

4. Have the authors made all data underlying the findings in their manuscript fully available (please refer to the Data Availability Statement at the start of the manuscript PDF file)?

Reviewer #2: No

Reviewer #3: Yes

Reviewer #4: No

5. Is the manuscript presented in an intelligible fashion and written in standard English?

Reviewer #2: Yes

Reviewer #3: Yes

Reviewer #4: Yes

6. Review Comments to the Author

Reviewer #2: No additional comments.

Reviewer #3: The authors have adequately addressed my comments.

Reviewer #4: General

The general presentation of the manuscript has improved.

Results

A few things need to be done in the results section. Currently the tables are not easy to read.

-Table 2: Are the values in the YES column in percentages, if so it needs to be clear in the table.

-Tables 3 & 4: The authors did a good job with regression analysis running different models. Which model did you decide to use for your results and why? Did you do a model fit and which type did you use to assess. To make the tables easier to read, please present only the results from the model with the best fit and give a brief statement of the process.

Discussion

The most important or main results are best presented in the first paragraph in 3-4 lines at most, to give readers a quick idea of what is to follow.

7. PLOS authors have the option to publish the peer review history of their article (what does this mean?). If published, this will include your full peer review and any attached files.

**Do you want your identity to be public for this peer review?** For information about this choice, including consent withdrawal, please see our Privacy Policy.

Reviewer #2: No

Reviewer #3: No

Reviewer #4: No

---

## [Decision Letter · Decision Letter 2]

17 Mar 2023

PGPH-D-22-00694R2

Identity and COVID-19 related symptoms in Canada: Gender, ethnicity, and minority status

Dear Dr. Yaya,

Thank you for submitting your manuscript to PLOS Global Public Health. After careful consideration, we feel that it has merit but does not fully meet PLOS Global Public Health’s publication criteria as it currently stands. Therefore, we invite you to submit a revised version of the manuscript that addresses the points raised during the review process.

We look forward to receiving your revised manuscript.

Kind regards,

Daniel Kim, M.D., Dr.P.H.

Academic Editor

Journal Requirements:

2. Your manuscript is missing the following sections: Introduction. Please ensure these are present, and in the correct order, and that any references to subheadings in your main text are correct. An outline of the required sections can be consulted in our submission guidelines here:

https://journals.plos.org/globalpublichealth/s/submission-guidelines#loc-parts-of-a-submission

Additional Editor Comments (if provided):

For Figures 1 and 2, please use better resolution graphs and also display 95% confidence intervals for the prevalence estimates.

Reviewers' comments:

Reviewer's Responses to Questions

**Comments to the Author**

1. If the authors have adequately addressed your comments raised in a previous round of review and you feel that this manuscript is now acceptable for publication, you may indicate that here to bypass the “Comments to the Author” section, enter your conflict of interest statement in the “Confidential to Editor” section, and submit your "Accept" recommendation.

Reviewer #4: All comments have been addressed

2. Does this manuscript meet PLOS Global Public Health’s publication criteria? Is the manuscript technically sound, and do the data support the conclusions? The manuscript must describe methodologically and ethically rigorous research with conclusions that are appropriately drawn based on the data presented.

Reviewer #4: Yes

3. Has the statistical analysis been performed appropriately and rigorously?

Reviewer #4: Yes

4. Have the authors made all data underlying the findings in their manuscript fully available (please refer to the Data Availability Statement at the start of the manuscript PDF file)?

Reviewer #4: No

5. Is the manuscript presented in an intelligible fashion and written in standard English?

Reviewer #4: Yes

6. Review Comments to the Author

Reviewer #4: Thank you for the updates and changes to the manuscript.

7. PLOS authors have the option to publish the peer review history of their article (what does this mean?). If published, this will include your full peer review and any attached files.

**Do you want your identity to be public for this peer review?** For information about this choice, including consent withdrawal, please see our Privacy Policy.

Reviewer #4: No

---

## [Editor Report · Decision Letter 3]

18 Apr 2023

Identity and COVID-19 related symptoms in Canada: Gender, ethnicity, and minority status

PGPH-D-22-00694R3

Dear Dr. Yaya,

We are pleased to inform you that your manuscript 'Identity and COVID-19 related symptoms in Canada: Gender, ethnicity, and minority status' has been provisionally accepted for publication in PLOS Global Public Health.

Best regards,

Daniel Kim, M.D., Dr.P.H.

Academic Editor